# Reference values of urinary metabolites of organophosphate in healthy Iranian adults

Rosa Haghshenas[1☯], Zahra Soleimani[1,2,3☯], Yosef Farzi[1], Amirmohammad Khalaji[1], Amir Hossein Behnoush[1], Ali Taherkhani[4], Ashkan Pourabhari Langroudi[1], Shahedeh Seyfi[1], Nasim Hadian[1], Mohammadhassan Momayyezi[5], Samaneh Mozafari[5], Samaneh Abolli[2,4], Kazem Naddafi[2,4], Masud Yunesian[4,6], Alireza Mesdaghinia[4,7]*, Farshad Farzadfar[1,8‡]

1 Non-Communicable Diseases Research Center, Endocrinology and Metabolism Population Sciences Institute, Tehran University of Medical Sciences, Tehran, Iran, 2 Center for Air Pollution Research (CAPR), Institute for Environmental Research (IER), Tehran University of Medical Sciences, Tehran, Iran, 3 Department of Environmental Health Engineering, School of Public Health, Semnan University of Medical Sciences, Semnan, Iran 4 Department of Environmental Health Engineering, School of Public Health, Tehran University of Medical Sciences. Tehran, Iran, 5 Environmental Health Expert, Provincial Health Center, Shahid Sadougi University of Medical Sciences, Yazd, Iran, 6 Department of Research Methodology and Data Analysis, Institute for Environmental Research (IER), Tehran University of Medical Sciences, Tehran, Iran, 7 Center for Water Quality Research, Institute for Environmental Research, Tehran University of Medical Sciences, Tehran, Iran, 8 Endocrinology and Metabolism Research Center, Endocrinology and Metabolism Clinical Sciences Institute, Tehran University of Medical Sciences, Tehran, Iran

☯ These authors contributed equally to this work.
‡ FF also contributed equally to this work.
* mesdaghinia@tums.ac.ir

## Abstract

Organophosphorus pesticides are widely used in agriculture in Iran; we evaluated exposure to these pesticides among Iranian adults. Pesticide-specific urinary metabolites were used as biomarkers for exposure to various pesticides, including organophosphorus insecticides. The aim of the study was to estimate reference values (RV95) and their relationships with the measured factors. We used the 95th percentile as the basis for deriving these reference values. The analysis included descriptive statistics and multiple linear regression, conducted using Python software. We measured metabolites for Chlorpyrifos (TCP: 2-isopropyl-4-methyl-6-hydroxypyrimidine), Diazinon (IMPY: 2-isopropyl-4-methyl-6-hydroxypyrimidine), and Malathion (Malathion dicarboxylic acid) in 490 healthy Iranian adults. Additionally, we recorded age, gender, wealth index, and body composition parameters including body fat, muscle mass, visceral fat, and BMI. Fasting urine sampling, along with body composition and demographic measurements, were conducted. Urine samples were subsequently analyzed. The Chlorpyrifos, Diazinon, and Malathion Reference Value (RV95) levels ranged from ND-24.9 µg/L (RV95: 2.8 µg/L, 2.9 µg/gcrt), ND-64.36 µg/L (RV95: 8.6 µg/L, 9.3 µg/gcrt), and ND-47.69 µg/L (RV95: 9.8 µg/L, 8.2 µg/gcrt), respectively. Diazinon (IMPY) and Malathion (Malathion dicarboxylic acid) showed no significant relationship between their urinary levels and demographic features. However, visceral

**Data availability statement:** All relevant data are within the paper and its Supporting Information files.

**Funding:** This research was funded by Grant 964655 from the National Institute for Medical Research Development (NIMAD) Iran.

**Competing interests:** No authors have competing interests Enter: The authors have declared that no competing interests exist.

fat percentage had a significant inverse correlation with urinary levels of Chlorpyrifos (TCP) (P = 0.038). Other factors such as age, sex, visceral fat, BMI, and wealth index showed no significant relationship with urinary levels (P > 0.05). Non-zero levels were found in 98.8% of adults' urine samples for this metabolite. The reference value of this pesticide metabolite in urine could be helpful for policymakers in assessing the level of exposure among Iranians.

## Introduction

Over the past century, population growth, urbanization, and industrial and agricultural advancements have increased exposure to heavy metals, pesticides, and polycyclic aromatic hydrocarbons [1]. Organophosphate pesticides are highly toxic chemicals that inhibit the enzyme acetylcholinesterase (AChE), leading to an accumulation of acetylcholine in the nervous system. This accumulation can cause severe health issues, including respiratory failure, seizures, and long-term neurological impairments, such as memory loss and mood disorders [1–3].

In Iran, the annual use of pesticides in agriculture has continued to rise, largely due to the expansion of agricultural activities across urban and rural areas [2–5]. Vulnerable groups, particularly agricultural workers and children, face heightened risks due to occupational exposure and environmental contamination [6,7]. As pesticide use in agriculture has risen, so has human exposure to these chemicals, which occurs not only through food consumption but also through the inhalation of contaminated dust and air [8–10]. The ecological effects are also alarming, as these pesticides can leach into ecosystems, harming non-target species like aquatic life and disrupting food webs [11].

There is growing concern about the overuse of pesticides in Iran, which has been linked to several public health issues in recent years [4]. Diazinon, in particular, has been identified as a significant pesticide residue in fruits and vegetables, often exceeding safety limits in the country. Chlorpyrifos, while approved for agricultural use, is closely monitored due to its toxicity. Regulatory efforts are underway to manage pesticide use more safely, but gaps remain in comprehensive data on pesticide residues in food crops, highlighting the need for improved monitoring and regulation to protect public health and the environment [5–7,9].

Human biomonitoring (HBM) involves collecting biological samples, such as blood, urine, and breast milk, to analyze pollutant concentrations and their metabolites [12,13]. This process determines human exposure to chemicals at both occupational and environmental levels, enabling corrective actions if necessary, and reveals differences in chemical distribution across countries and regions [14–16]. Urine is the most utilized matrix in biomonitoring studies due to its abundance, ease of collection, and ability to improve analytical quantification levels for chemicals with a low response factor in chromatography [17,18].

HBM data from nationally representative surveys are considered the best source for calculating reference values (RV95s) for environmental chemicals in human

biomaterials [13]. An RV95 represents the upper limit of the general population's background exposure to a substance at a specific period [19].

Although researchers commonly utilize reported RVs to interpret data, country-specific variables such as lifestyle factors, food consumption patterns, and levels of ambient pollution can result in significant differences in RVs among populations. Therefore, RV95s must be established at national or regional levels to accurately reflect local conditions.

The German HBM Commission defines the reference value, RV95, as the 95th percentile of the substance of interest at a specific time point, rounded off within its 95% confidence interval (CI) [20], which aligns with the International Union of Pure and Applied Chemistry (IUPAC) guidelines [21]. The selection of the 95th percentile and 95% CI is motivated by the convention in hypothesis testing, where the 5% most extreme sample values indicate unusual values [20].

Several countries, including Canada, Germany, and Spain, have successfully conducted HBM studies to monitor exposure to chemical compounds and calculate RV95s [20,22–24]. However, in Iran, despite a sharp increase in pesticide use over the past decade—particularly the use of organophosphorus pesticides in agriculture—there is a notable absence of national biomonitoring data and no reported RV95 values specific to the Iranian population. This gap raises significant concerns about the toxicological and health impacts of pesticide exposure in Iran.

This study aims to address this critical gap by calculating RV95s for pesticide metabolites using data from an HBM study that analyzed urine samples and examined key demographic factors in Iran.

## Materials and methods

### Sampling

In this survey, the sample size and sampling locations were based on the protocol study that we have widely discussed in protocol article [25]. Sampling was done from 1/5/2019–15/3/2021 on Iranian adults aged 25 and above. Overall, 660 households participated in six provinces (Fig 1) [25], after data cleaning, 490 samples were used for statistical analysis in this research. Data collection was done in confidentially [25].

The sampling design was developed based on a calculated sample size of 383, assuming a 50% baseline prevalence for urinary toxic metal(loid) levels, as no prior national surveys had assessed these parameters. Using Cochran's formula for large populations [26] and accounting for a 1.5 design effect [27] and an 85% response rate [28], the sample size was adjusted to 660 across 132 clusters. Clusters were chosen via model-based clustering [29,30], allowing us to organize Iranian provinces into six super-clusters (SCs) based on covariates such as particulate matter (PM10), pesticide use, and rates of cancer and chronic respiratory diseases. The capitals of these provinces—Ardabil, Kermanshah, Mashhad, Zahedan, Tehran, and Yazd—served as essential sampling units, with cluster coordinates within urban zones generated for systematic sampling [31]. Table 1 and S1 Table provides details on each SC, cluster, and assigned sample size and inclusion and exclusion criteria.

After Recruitment people addresses, randomly selected according to the sampling method in protocol article, two copies of the consent form were filled out and signed by each participant, (One copy was given to the participant and one archived). Then two questionnaires (Demographic, Household Assets) all valid and reliable filled, the demographic questionnaire collected information about age, gender, education, and career. In addition, Anthropometry gathered the anthropometric indices and body composition that measured by portable body composition scale (Omron, BF 511) with 100-g precision. The mobile body composition scale is an 8-sensor bioelectrical impedance analyzer that can report body fat (in %), skeletal muscle (in %), and visceral fat. The gender, age, and height of the participants entered manually. We measure of metabolite malathion, diazinon and chlorpyrifos Because these three compounds are used more in Iran, we chose them [25]. This research with this code of ethics IR.NIMAD. REC.1397.310 is in accordance with the ethical principles and national norms and standards of conducting medical research in Iran.

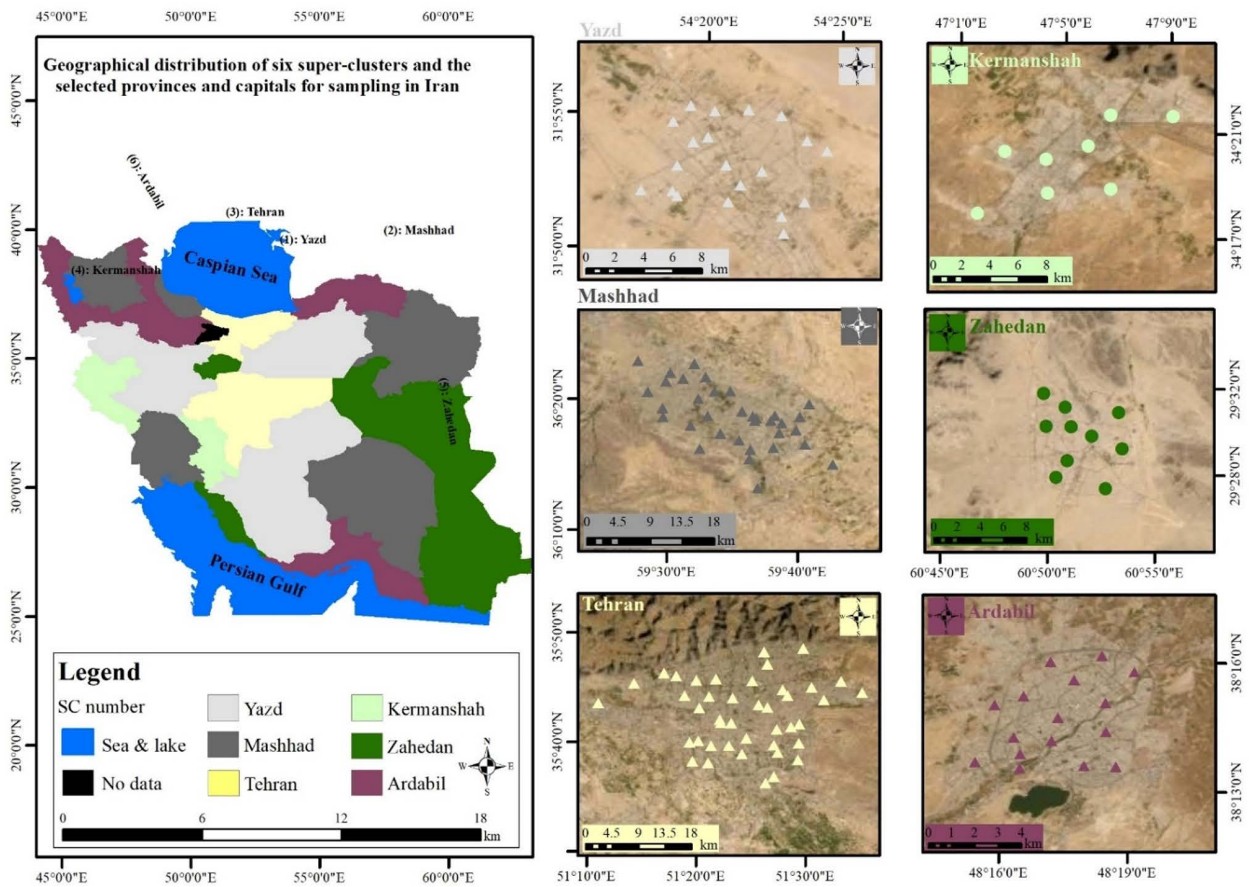

**Fig 1. Geographical distribution of the six super-clusters (SCs) and the selected provinces and capitals for sampling in Iran [25].**

**Table 1. Inclusion and exclusion criteria in this study [25].**

| Inclusion | Exclusion | Explain |
|---|---|---|
| Age 25 or over | Pregnant | – |
| Not smoker and contact with smokers | contact with smoker in home and work place | – |
| Individuals should be apparently healthy. | Gets sick | – |
| Individuals should not take any medication within the last month. | Gets sick | – |
| Individuals should not have a history of any chemo-therapy or radiotherapy throughout their lifetime. | Gets sick | – |
| Occupational exposure is not allowed | Have an occupational exposure | Their occupation should not include the following items; Casting, Metallurgy, Welding, and Soldering Rubber, asphalt, and coke industries Pesticide manufacturing industries Agriculture and Greenhouse Cooking (in large restaurants) |

## Sampling urine

For sampling and measurement of pollutants in urine, we measured metabolized pesticides in urine samples. The urine sample of 300 ml fasting urine was collected from participants. The samples were stored at 4°C in cooling boxes during transport to ensure preservation until laboratory processing. The samples were then separated into aliquots and stored at −20°C for a few days before being transferred to the laboratory or stored in −80°C units for long-term storage. Participants received explicit instructions in the form of a leaflet on how to collect the urine sample according to the study's Standard Operating Procedure. Female participants were given specific instructions on washing before sampling during menstruation [32].

## Sample preparation and analysis

The target pesticides were analyzed in urine samples following enzymatic deconjugation and solid-phase extraction, similar to the method described by the CDC [33]. Briefly, 0.5 mL of urine was transferred into a 15-mL polypropylene (PP) tube, and 1 ng of each labeled internal standard mixture was added. The urine samples were mixed with 400 μL of 0.2 M sodium acetate buffer containing 745 units/mL of β-glucuronidase and 56 units/mL of sulfatase, then incubated at 37°C for at least 6 hours. The samples were subsequently passed through Oasis® HLB 3 mL solid-phase extraction cartridges that had been conditioned with 2 mL of acetone and 2 mL of 1% acetic acid in water. After loading the samples, the cartridges were washed with acetic acid/methanol/water (1:5:94, v/v/v) and vacuum dried for 5 minutes. Analytes were recovered by elusion with 3 mL of acetone followed by 3 mL of hexane. The combined eluates were evaporated to dryness under nitrogen, re-dissolved in 100 μL of acetonitrile in water (1:1, v/v), and transferred into glass vial inserts for instrumental analysis [34].

## Quality Assurance and Quality Control (QA & QC)

Standard solutions were prepared using Pesticide Mix 1510 (Sigma Aldrich Company) at a concentration of 50 ppm, which included the pesticides Diazinon, Malathion, and Chlorpyrifos. From this mix, standards at concentrations of 20, 50, 100, and 200 ppb were prepared and analyzed in the LC-MS device in the same volume as the samples. Isotopically labeled Dialkyl Phosphates, including Diethyl Phosphate (DEP)-d10 and Dimethyl Phosphate (DMP)-d6, were used as internal standards [35,36].

This study employed a rigorous methodology to validate calibration curves and ensure reliable results. Calibration standards were created at 20, 50, 100, and 200 ppb, and these were analyzed using the LC-MS device. The calibration curve was validated through linear regression analysis, targeting a correlation coefficient (r) greater than 0.995. Acceptance criteria required back-calculated concentrations to be within ±15% of nominal values, with the exception of the lower limit of quantification (LLOQ), which was allowed a deviation of ±20% (Table 2).

Additionally, QC samples were prepared by spiking a blank matrix to continuously monitor method performance. Accuracy and precision were assessed according to the same stringent criteria applied to the calibration standards. These comprehensive QA & QC measures ensured the robustness and reliability of the analytical method throughout the study.

**Table 2. LOD and LOQ for the compounds in the pesticides.**

| Compound | LOD (ppb) | LOQ (ppb) | RSD | R2 | Recovery (%) |
|---|---|---|---|---|---|
| Chlorpyrifos (TCP) | 0.6 | 0.96 | 6.78 | 0.997 | 79 |
| Diazinon (IPMP) | 0.5 | 1.65 | 9.7 | 0.998 | 82 |
| Malathion (MALOXONE) | 1 | 3.3 | 10.9 | 0.998 | 80 |

## Statistical analysis

All analyses were conducted using Python, version 3.7.8 (64-bit). Data were tested for normality prior to analysis using the Kolmogorov–Smirnov test. Descriptive analysis was performed to calculate the mean, standard deviation, and confidence interval of the mean for the pesticide levels. These results were analyzed by station and by sex. Pesticide values below the limit of detection (LOD) were replaced by LOD/√2 [37], a method that has shown desirable results in pesticide bio-marker analysis [38].

A descriptive table of the wealth index based on stations was then created. Physical measurements of the participants, including sex, visceral fat, muscle percentage, body fat, and body mass index, were analyzed across different age groups. Pesticide levels are presented both in µg/L and µg/g creatinine (converted to per gram of creatinine from micrograms per liter using the creatinine concentration measured in each participant's urine).

Using data from asset questionnaires, participants were classified into five distinct wealth groups based on their wealth index. This categorization was achieved by applying principal component analysis to the assets of all households in the study, resulting in five wealth groups based on the derived index. The detailed methodology for determining the wealth index for each family is thoroughly described in Hjelm et al. (2017) [39].

Potential relationships between physical measurements and pesticide levels in urine were analyzed using multivariate linear regression analysis. Before performing the regression, the distribution of each pesticide was tested for normality. The Kolmogorov–Smirnov test was applied to each variable to assess normality. If the data did not follow a normal distribution, a log transformation was considered for the regression. Multivariate linear regression analysis was then performed for each pesticide, with variables including the physical characteristics of the participants: sex, age, visceral fat, muscle percentage, body fat, and body mass index.

## Results

### Baseline characteristics

Of the 490 participants from six stations (Mashhad, Yazd, Ardebil, Kermanshah, Zahedan, and Tehran), Kermanshah had the lowest and Mashhad the highest population in this study. Table 3 shows the study population divided into five quintiles: the Q1 quintile represents those at the lower end of the income/wealth spectrum, and the Q5 quintile represents those at the top. While participants in Tehran were predominantly in the Q5 quintile, Ardebil had the highest population in the Q1 quintile. The largest proportion of participants in Mashhad fell into the Q3 quintile.

Table 4 presents the characteristics of the study participants. The majority were female (58.09%); however, in the age group of 55–65, the gender distribution was equal. Most participants were younger than 45 years old, with the 35–45 age group being the largest.

**Table 3. Economic status of people participating in the study.**

| Station | Q1 | Q2 | Q3 | Q4 | Q5 |
|---|---|---|---|---|---|
| Mashhad | 11 | 25 | 34 | 30 | 25 |
| Yazd | 19 | 13 | 10 | 17 | 18 |
| Ardebil | 41 | 29 | 11 | 7 | 3 |
| Kermanshah | 7 | 7 | 13 | 16 | 4 |
| Zahedan | 12 | 16 | 16 | 2 | 4 |
| Tehran | 5 | 8 | 14 | 28 | 45 |
| Total | 95 | 98 | 98 | 100 | 99 |

**Table 4. Baseline demographics of participants based on different age groups.**

| Age group | | 25–35 | | 35–45 | | 45–55 | | 55–65 | | >65 | | Total | | |
|---|---|---|---|---|---|---|---|---|---|---|---|---|---|---|
| | | Female | Male | Female | Male | Female | Male | Female | Male | Female | Male | Female) | Male | Both |
| Height (cm) | Mean±SD | 160.89±6.21 | 174.43±6.79 | 159.88±6.42 | 174.99±6.52 | 158.17±6.31 | 170.58±7.78 | 155.11±6.02 | 170.94±6.21 | 151.67±3.08 | 161.67±12.50 | 159.43±6.50 | 173.10±7.72 | 164.02±9.47 |
| P_value | | <0.001 | | <0.001 | | <0.001 | | <0.001 | | 0.09 | | <0.001 | | |
| Weight (kg) | Mean±SD | 66.2±11.76 | 78.07±15.22 | 71.92±12.65 | 83.92±14.31 | 73.83±13.44 | 75.67±13.85 | 70.00±12.53 | 81.91±12.29 | 67.92±9.05 | 66.13±18.69 | 70.11±12.72 | 79.63±15.02 | 73.30±14.25 |
| P_value | | <0.001 | | <0.001 | | 0.58 | | 0.01 | | 0.84 | | <0.001 | | |
| Body mass index (kg/m²) | Mean±SD | 25.6±4.7 | 26.83±4.74 | 27.99±4.27 | 27.81±4.17 | 29.87±4.51 | 26.33±3.42 | 29.19±5.39 | 28.73±4.73 | 29.53±3.95 | 24.75±4.37 | 27.61±4.78 | 27.19±4.35 | 27.47±4.64 |
| P_value | | 0.17 | | 0.83 | | <0.001 | | 0.81 | | 0.07 | | 0.4 | | |
| Body fat (%) | Mean±SD | 37.99±7.75 | 26.75±8.58 | 40.95±5.80 | 28.22±7.15 | 42.74±6.73 | 25.47±5.61 | 41.91±7.71 | 29.48±8.50 | 43.23±5.47 | 23.70±9.96 | 40.36±7.02 | 27.14±7.75 | 35.97±9.57 |
| P_value | | <0.001 | | <0.001 | | <0.001 | | <0.001 | | <0.001 | | <0.001 | | |
| Body muscle (%) | Mean±SD | 25.27±2.44 | 31.80±5.25 | 24.90±2.44 | 32.16±4.02 | 24.72±3.22 | 33.11±3.41 | 24.58±3.83 | 30.7±4.5 | 24.33±2.39 | 32.05±3.27 | 24.97±2.68 | 32±4.41 | 27.37±4.74 |
| P_value | | <0.001 | | <0.001 | | <0.001 | | <0.001 | | <0.001 | | <0.001 | | |
| Visceral Fat (%) | Mean±SD | 5.76±1.78 | 7.96±3.76 | 7.52±2.08 | 9.68±3.98 | 8.92±2.16 | 9.67±3.16 | 9.39±2.35 | 10.94±3.61 | 11.67±2.07 | 8.33±3.50 | 7.38±2.45 | 9.17±3.79 | 7.98±3.08 |
| P_value | | <0.001 | | <0.001 | | <0.001 | | 0.13 | | 0.07 | | <0.001 | | |
| Resting Metabolism (Cal) | Mean±SD | 1365.31±132.20 | 1671.17±34.72 | 1404.44±145.60 | 1709.76±02.57 | 1440.59±76.86 | 1646.25±84.13 | 1392.03±72.34 | 1666.65±36.69 | 1331.00±86.69 | 1466.83±270.3 | 1395.31±150.22 | 1669.18±210.64 | 1487.25±215.78 |
| P_value | | <0.001 | | <0.001 | | <0.001 | | <0.001 | | 0.27 | | <0.001 | | |

## Anthropometric indices of men and women

Men were generally taller and weighed more than women, while women had higher BMI and body fat percentages. Detailed statistics are provided in Table 4. Body muscle percentage was higher in men (32.00±4.41 vs. 24.97±2.68), and men also had a higher visceral fat percentage compared to women (9.17±3.79 vs. 7.38±2.45). In addition, men showed a higher resting metabolic rate (1669.18±210.64 Cal vs. 1395.31±150.22 Cal). For height, weight, body fat percentage, and resting metabolism, there are significant differences between males and females across most age groups, as indicated by p-values less than 0.001. This suggests that these physical measurements are consistently different between genders. However, for body mass index (BMI), significant differences are only observed in the 45–55 age group, with p-values indicating no significant differences in other age groups. Visceral fat percentage shows significant gender differences in some age groups but not others. The only instance where no significant gender difference is observed for height is in the >65 age group (p-value=0.09), and for weight, there is no significant difference in the 55–65 age group (p-value=0.84).

## Urine pesticide levels in the Iranian population

The adjusted and unadjusted urine levels of Chlorpyrifos (TCP), Diazinon (IMPy), and Malathion (Malathion dicarboxylic acid), categorized by participants' sex, are available in Tables 5 and 6. The mean and RV95 urine concentration of Chlorpyrifos (TCP) were similar between men and women, with mean values of 0.768±0.173 µg/L for women and 0.718±0.295 µg/L for men, and RV95 values of 3.0 µg/L for women and 2.6 µg/L for men (Table 5). However, the creatinine-corrected concentration was higher in men than in women. Adjusted values increased in men (0.888±0.555 µg/gcrt) and decreased in women (0.754±0.181 µg/gcrt), widening the difference between sexes; RV95 increased in women (RV95: 3 µg/gcrt) and decreased in men (RV95: 2.1 µg/gcrt) (Table 5). Among super-clusters, Yazd had the lowest unadjusted (0.05±0.02 µg/L) and adjusted (0.06±0.03 µg/gcrt) Chlorpyrifos (TPC) levels, while Kermanshah had the highest

**Table 5. Level of pesticides in urine of the Iranian population.**

| Asset | Type | Population | Mean±SE | P_value | 95% CI | P5 | P10 | P50 | P90 | P95 | RV95 |
|---|---|---|---|---|---|---|---|---|---|---|---|
| Chlorpyrifos (TCP) | (µg/L) | Female | 0.768±0.173 | 0.65 | 0.428–1.11 | 0.004 | 0.004 | 0.115 | 1.18 | 3.00 | 3.0 |
| | | Male | 0.718±0.295 | | 0.130–1.30 | 0.004 | 0.004 | 0.085 | 0.996 | 2.57 | 2.6 |
| | | Both | 0.75±0.15 | | 0.46–1.05 | 0.004 | 0.004 | 0.106 | 1.13 | 2.82 | 2.8 |
| | (µg/g crt) | Female | 0.754±0.181 | 0.98 | 0.398–1.11 | 0.002 | 0.003 | 0.114 | 1.18 | 3.28 | 3.3 |
| | | Male | 0.888±0.555 | | −0.216–1.99 | 0.002 | 0.002 | 0.055 | 1.09 | 2.06 | 2.1 |
| | | Both | 0.79±0.20 | | 0.39–1.19 | 0.002 | 0.002 | 0.092 | 1.16 | 2.88 | 2.9 |
| Diazinon (IPMP) | (µg/L) | Female | 2.67±0.231 | 0.07 | 2.22–3.13 | 0.148 | 0.382 | 1.80 | 5.68 | 7.78 | 7.8 |
| | | Male | 3.77±0.774 | | 2.23–5.31 | 0.131 | 0.393 | 2.18 | 7.40 | 9.38 | 9.4 |
| | | Both | 2.98±0.27 | | 2.44–3.51 | 0.137 | 0.387 | 1.92 | 5.93 | 8.60 | 8.6 |
| | (µg/g crt) | Female | 2.83±0.283 | 0.94 | 2.27–3.39 | 0.195 | 0.378 | 1.55 | 6.71 | 9.77 | 9.8 |
| | | Male | 2.79±0.560 | | 1.67–3.90 | 0.194 | 0.259 | 1.50 | 6.29 | 8.51 | 8.5 |
| | | Both | 2.82±0.26 | | 2.31–3.32 | 0.194 | 0.328 | 1.52 | 6.70 | 9.28 | 9.3 |
| Malathion | (µg/L) | Female | 2.57±0.232 | 0.03 | 2.11–3.03 | 0.007 | 0.189 | 1.53 | 5.84 | 9.16 | 9.2 |
| | | Male | 3.84±0.682 | | 2.49–5.20 | 0.121 | 0.448 | 1.92 | 8.07 | 14.4 | 14 |
| | | Both | 2.92±0.25 | | 2.42–3.42 | 0.007 | 0.242 | 1.62 | 6.61 | 9.84 | 9.8 |
| | (µg/g crt) | Female | 2.33±0.209 | 0.3 | 1.92–2.75 | 0.014 | 0.236 | 1.43 | 5.11 | 8.03 | 8.0 |
| | | Male | 2.82±0.525 | | 1.78–3.87 | 0.072 | 0.314 | 1.27 | 5.34 | 10.1 | 10.1 |
| | | Both | 2.47±0.21 | | 2.06–2.89 | 0.016 | 0.260 | 1.36 | 5.19 | 8.19 | 8.2 |

Data are presented as mean±SE. Abbreviations: SE, standard Error; CI, confidence interval.

**Table 6. Concentrations of pesticides in different super-clusters.**

| Station | Unite | Gender | Chlorpyrifos (TCP) Mean±SE | P_value | Diazinon (IPMP) Mean±SE | P_value | Malathion Mean±SE | P_value |
|---|---|---|---|---|---|---|---|---|
| Ardebil | (µg/L) | Female | 0.285±0.180 | 0.33 | 3.02±0.716 | 0.72 | 2.63±0.691 | 0.69 |
| | | Male | 0.608±0.334 | | 2.68±0.598 | | 2.25±0.634 | |
| | | Both | 0.43±0.18 | | 2.87±0.47 | | 2.45±0.47 | |
| | (µg/g crt) | Female | 0.116±0.072 | 0.17 | 2.09±0.491 | 0.98 | 1.62±0.293 | 0.46 |
| | | Male | | 0.596±0.388 | | 2.07±0.574 | | 1.31±0.284 |
| | | Both | 0.33±0.18 | | 2.09±0.37 | | 1.48±0.20 | |
| Tehran | (µg/L) | Female | 0.269±0.215 | 0.47 | 0.962±0.183 | <0.001 | 1.64±0.586 | 0.1 |
| | | Male | 0.023±0.007 | | 3.59±0.846 | | 3.78±1.34 | |
| | | Both | 0.18±0.14 | | 1.93±0.37 | | 2.43±0.63 | |
| | µg/g crt) | Female | 0.162±0.112 | 0.4 | 0.772±0.104 | <0.001 | 1.20±0.328 | 0.06 |
| | | Male | | 0.022±0.008 | | 3.10±0.952 | | 2.65±0.774 |
| | | Both | 0.11±0.07 | | 1.69±0.41 | | 1.78±0.37 | |
| Zahedan | (µg/L) | Female | 2.14±0.927 | 0.15 | 2.73±0.665 | 0.74 | 3.53±0.802 | 0.23 |
| | | Male | 0.540±0.467 | | 3.21±1.56 | | 5.54±1.60 | |
| | | Both | 1.72±0.70 | | 2.86±0.63 | | 4.06±0.73 | |
| | µg/g crt) | Female | 1.77±0.876 | 0.18 | 2.86±0.798 | 0.45 | 2.61±0.453 | 0.26 |
| | | Male | | 0.253±0.176 | | 1.81±0.581 | | 3.66±0.841 |
| | | Both | 1.37±0.65 | | 2.58±0.61 | | 2.89±0.40 | |
| Mashhad | (µg/L) | Female | 0.384±0.031 | 0.19 | 2.73±0.248 | 0.43 | 2.39±0.270 | 0.91 |
| | | Male | 0.478±0.074 | | 2.37±0.132 | | 2.33±0.492 | |
| | | Both | 0.41±0.03 | | 2.64±0.19 | | 2.38±0.24 | |
| | µg/g crt) | Female | 0.441±0.045 | 0.71 | 3.21±0.314 | 0.03 | 2.68±0.322 | 0.72 |
| | | Male | | 0.405±0.088 | | 1.87±0.194 | | 2.40±0.962 |
| | | Both | 0.43±0.04 | | 2.91±0.25 | | 2.62±0.33 | |
| Kermanshah | (µg/L) | Female | 1.69±0.639 | 0.61 | 4.52±1.24 | 0.09 | 3.27±0.856 | 0.08 |
| | | Male | 5.33±3.94 | | 13.5±10.2 | | 10.1±7.62 | |
| | | Both | 2.32±0.85 | | 6.07±2.00 | | 4.44±1.47 | |
| | µg/g crt) | Female | 1.67±0.702 | 0.11 | 3.80±1.25 | 0.15 | 2.71±0.868 | 0.15 |
| | | Male | | 8.34±7.51 | | 9.70±6.69 | | 6.98±4.97 |
| | | Both | 2.81±1.39 | | 4.81±1.53 | | 3.44±1.10 | |
| Yazd | (µg/L) | Female | 0.028±0.014 | 0.02 | 2.16±0.549 | 0.87 | 2.22±0.843 | 0.07 |
| | | Male | 0.138±0.082 | | 5.70±1.43 | | 5.68±1.68 | |
| | | Both | 0.05±0.02 | | 2.96±0.61 | | 3.01±0.80 | |
| | µg/g crt) | Female | 0.066±0.039 | 0.14 | 3.88±3.28 | 0.07 | 1.21±0.623 | 0.08 |
| | | Male | | 0.045±0.043 | | 3.06±0.841 | | 3.34±0.553 |
| | | Both | 0.06±0.03 | | 3.63±2.26 | | 1.85±0.55 | |
| Iran | (µg/L) | Both | 0.75±0.15 | 0.98 | 2.98±0.27 | 0.07 | 2.92±0.25 | 0.03 |
| | µg/g crt) | Both | 0.79±0.20 | 0.65 | 2.82±0.26 | 0.94 | 2.47±0.21 | 0.3 |

unadjusted (2.32±0.85 µg/L) and adjusted (2.81±1.39 µg/gcrt) levels (Table 6). The Concentrations of pesticides in different super-clusters are shown in Table 6. For Chlorpyrifos (TCP), most p-values are not statistically significant, indicating that the observed differences between genders are likely due to random chance. However, in Yazd, the p-value is 0.02 for females in µg/L, suggesting a statistically significant difference. For Diazinon (IPMP), the p-value is less than

0.001 in Tehran, indicating a significant difference between genders. Malathion shows significant differences in Iran overall (p-value = 0.03) and in Yazd (p-value = 0.07), suggesting that gender differences in Malathion concentrations are unlikely to occur by chance in these areas (Table 6). Overall, these p-values help assess whether observed differences in pesticide exposure are due to random variation or reflect real gender differences.

For Diazinon (IMPy), mean and RV95 urine concentrations were higher in men than women (3.77 ± 0.774 µg/L [RV95: 9.4 µg/L] vs. 2.67 ± 0.231 µg/L [RV95: 7.8 µg/L]) (Table 5). However, creatinine-corrected levels and RV95 values showed slightly different trends, with women exhibiting higher concentrations (2.83 ± 0.283 µg/gcrt [RV95: 9.8 µg/gcrt] vs. 2.79 ± 0.560 µg/gcrt [RV95: 8.5 µg/gcrt]). In men, adjusted Diazinon levels were lower than unadjusted levels (Table 5). Concentrations of Diazinon in the male population of Kermanshah were sparse, resulting in high SE values (unadjusted: 13.5 ± 10.2 µg/L; adjusted: 9.70 ± 6.69 µg/L). The lowest urine levels of Diazinon in the total population were found in Tehran (unadjusted: 1.93 ± 0.37 µg/L; adjusted: 1.69 ± 0.41 µg/gcrt) (Table 6).

For Malathion (Malathion dicarboxylic acid), men had higher mean and RV95 urine levels than women (3.84 ± 0.682 µg/L [RV95: 14 µg/L] vs. 2.57 ± 0.232 µg/L [RV95: 9.2 µg/L]). However, after adjusting for creatinine, the difference between men and women was greatly reduced (women: 2.33 ± 0.209 µg/gcrt [RV95: 8.0 µg/gcrt]; men: 2.82 ± 0.525 µg/gcrt [RV95: 10.1 µg/gcrt] (Table 5). Overall, as with Diazinon, adjusted Malathion levels decreased after creatinine adjustment (from 2.92 ± 0.25 µg/L to 2.47 ± 0.21 µg/gcrt). Among the total population, the urine concentration of Malathion (2.92 ± 0.25 µg/L) was higher than that of Chlorpyrifos (TPC) (0.75 ± 0.15 µg/L) but lower than that of Diazinon (2.98 ± 0.27 µg/L), as shown in Table 6 and Fig 2.

For Chlorpyrifos (TCP), the p-values are generally high (e.g., 0.65 for females in µg/L), indicating that the observed differences between genders are not statistically significant, suggesting that the data are consistent with the null hypothesis of no difference (Table 5). For Diazinon (IPMP), p-values are also high (e.g., 0.07 for females in µg/L) (Table 5), but this value is closer to the typical significance threshold of 0.05, suggesting some evidence against the null hypothesis, though not strong enough to reject it conclusively. Malathion shows a statistically significant difference between genders in µg/L (p-value = 0.03), indicating that the observed differences are unlikely to occur by chance, providing evidence against the null hypothesis (Table 5).

### The relationship between pesticide levels and demographic features

The relationship between creatinine-adjusted urinary levels of Chlorpyrifos (TCP), Diazinon (IMPy), and Malathion with demographic features is summarized in Table 7. Although no significant association was found between Diazinon or Malathion levels and demographic features, visceral fat percentage showed a significant negative correlation with Chlorpyrifos (TCP) levels (β = 0.927 [95% CI 0.862 to 0.996]; P = 0.038).

## Discussion

### Key findings and applications

The current study examined urinary pesticide concentrations, including creatinine-corrected levels, in relation to gender and geographical superclusters. The findings revealed that Kermanshah had the highest corrected urine levels of Chlorpyrifos, while Tehran had the lowest. Similarly, Kermanshah recorded the highest Malathion concentration among the six regions. In the case of Diazinon, the highest levels were observed in the Yazd cluster, with a corrected concentration of 5.19 ± 8.19 µg/g creatinine. The study also examined associations between demographic features and pesticide concentrations, with significant correlations found only between Chlorpyrifos levels, body fat, and BMI. The underlying mechanism for this association might relate to observations from animal studies, where Chlorpyrifos exposure was linked to dyslipidemia and obesity [40,41]. The biological cause may involve Chlorpyrifos promoting adipogenesis in 3T3-L1 adipocytes and fat accumulation, alongside inhibition of AMPK phosphorylation, thus encouraging expression of key adaptogenic regulators [42]. The lack of association between sex and age and pesticide levels suggests that the entire population is at risk of exposure

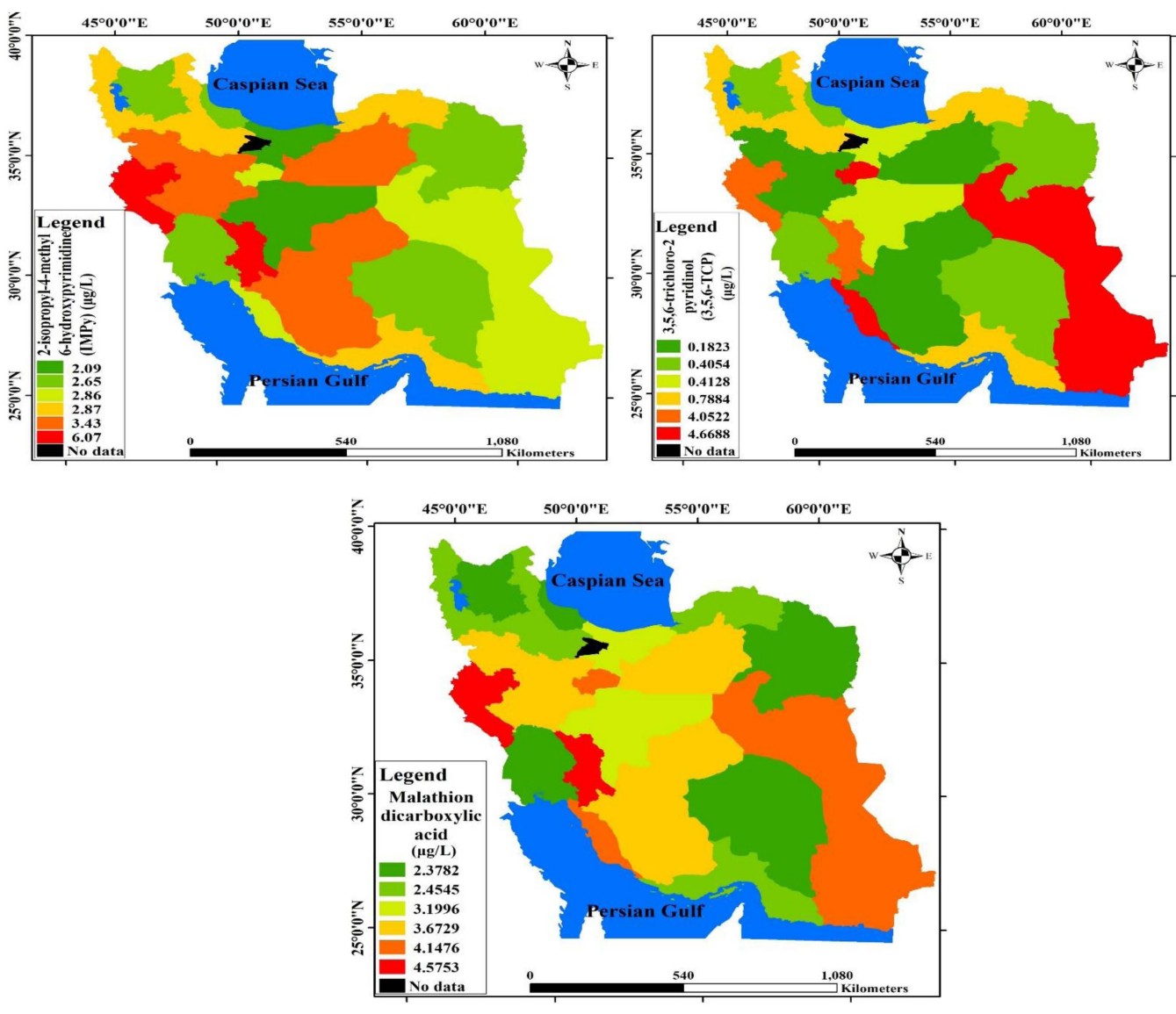

**Fig 2. Map of the state of pesticide metabolite concentrations measured urine in Iranian adults.**

to these chemicals, underscoring the need for inclusive policies in Iran. Furthermore, unlike many exposures, no correlation was found between wealth index and urine levels of Chlorpyrifos, Diazinon, and Malathion. Consequently, interventions should not be limited by socioeconomic status but should target the general population [43,44].

### Chlorpyrifos

Chlorpyrifos is a broad-spectrum organophosphate insecticide widely used globally [45], with neurotoxicity linked to inhibition of acetylcholinesterase [46]. This substance has been extensively discussed in the literature; Table 8 provides an expanded summary of studies reporting urinary chlorpyrifos metabolite levels across diverse populations and regions. Earlier research was concentrated in the U.S. during the 1990s and 2000s, while more recent studies are from Europe and Asia [47–59]. These newer studies report mean values ranging from 1.69 μg/L in Swedish adolescents to 7.35 μg/L in

**Table 7. Relationship between pesticide concentrations in urine and demographic features.**

| Variables | Chlorpyrifos (TCP) | | | Diazinon (IPMP) | | | Malathion | | |
|---|---|---|---|---|---|---|---|---|---|
| | β | 95% CI | P-value | β | 95% CI | P-value | β | 95% CI | P-value |
| Constant | 0.037 | 0.005–0.288 | 0.002 | 0.861 | 0.257–2.883 | 0.808 | 0.934 | 0.236–3.699 | 0.922 |
| Sex | 0.810 | 0.486–1.351 | 0.418 | 1.068 | 0.792–1.441 | 0.663 | 1.008 | 0.717–1.416 | 0.965 |
| Age | 0.998 | 0.984–1.013 | 0.823 | 1.000 | 0.992–1.009 | 0.931 | 0.992 | 0.982–1.002 | 0.107 |
| Visceral fat | **0.927** | **0.862–0.996** | **0.038** | 0.996 | 0.955–1.039 | 0.841 | 1.019 | 0.971–1.069 | 0.436 |
| Body muscle | 1.042 | 0.988–1.100 | 0.125 | 1.001 | 0.970–1.033 | 0.959 | 1.008 | 0.973–1.044 | 0.655 |
| Body fat | 1.022 | 0.988–1.057 | 0.202 | 1.001 | 0.982–1.021 | 0.885 | 0.997 | 0.975–1.019 | 0.786 |
| Body mass index | 1.026 | 0.973–1.083 | 0.337 | 1.007 | 0.976–1.040 | 0.642 | 1.010 | 0.974–1.046 | 0.595 |
| Wealth index | 1.043 | 0.934–1.164 | 0.449 | 0.985 | 0.924–1.050 | 0.648 | 0.971 | 0.903–1.045 | 0.433 |

CI, confidence interval.

In addition, a non-significant negative association was observed between Chlorpyrifos levels and sex (β = 0.810 [95% CI 0.486 to 1.351]; P = 0.418) and age (β = 0.998 [95% CI 0.984 to 1.013]; P = 0.823). Conversely, a non-significant positive relationship was noted between Chlorpyrifos levels and other demographic factors, including body muscle (β = 1.042 [95% CI 0.988 to 1.100]; P = 0.125), body fat (β = 1.022 [95% CI 0.988 to 1.057]; P = 0.202), BMI (β = 1.026 [95% CI 0.973 to 1.083]; P = 0.337), and wealth index (β = 1.043 [95% CI 0.934 to 1.164]; P = 0.449).

No significant associations were found between creatinine-adjusted urinary levels of Diazinon (IMPy) and demographic factors. However, a non-significant positive association was noted with sex (β = 1.068 [95% CI 0.792 to 1.441]; P = 0.663), body muscle (β = 1.001 [95% CI 0.970 to 1.033]; P = 0.959), body fat (β = 1.001 [95% CI 0.982 to 1.021]; P = 0.885), and BMI (β = 1.007 [95% CI 0.976 to 1.040]; P = 0.642). The relationships with age (β = 1.000 [95% CI 0.992 to 1.009]; P = 0.931), visceral fat (β = 0.996 [95% CI 0.955 to 1.039]; P = 0.841), and wealth index (β = 0.985 [95% CI 0.924 to 1.050]; P = 0.648) were negative but not significant.

Lastly, for adjusted Malathion levels, all associations with demographic features were non-significant. Positive relationships were noted with sex (β = 1.008 [95% CI 0.717 to 1.416]; P = 0.965), visceral fat (β = 1.019 [95% CI 0.971 to 1.069]; P = 0.436), body muscle (β = 1.008 [95% CI 0.973 to 1.044]; P = 0.655), and BMI (β = 1.010 [95% CI 0.974 to 1.046]; P = 0.595). Negative associations were found with age (β = 0.992 [95% CI 0.982 to 1.002]; P = 0.107), body fat (β = 0.997 [95% CI 0.975 to 1.019]; P = 0.786), and wealth index (β = 0.971 [95% CI 0.903 to 1.045]; P = 0.433).

Portuguese adults. Compared to these, our current study's mean adjusted value of 0.79 µg/g creatinine in Iranian adults is among the lowest reported. This contrast may reflect differences in pesticide regulation, dietary patterns, and environmental exposure. Notably, the Iranian values are also lower than the 95th percentile levels reported in recent HBM4EU-aligned studies, underscoring the value of establishing localized reference values to inform regional risk assessments and policy decisions [41,44].

Our results are also comparable to those from the European Human Biomonitoring Initiative (HBM4EU), which prioritizes pesticides for exposure monitoring [60]. According to HBM4EU data, Cyprus showed a high median (P50) TCPy urine level among children, while this was Israel for adults [61]. However, there has been a significant decline in pesticide exposure in European countries since 2016 [61]. Interestingly, HBM4EU studies found a positive correlation between higher education levels, lower BMI, and TCPy, contrary to our findings [49,62]. This difference may reflect Iran's limited dietary and exposure variance across educational levels. Additionally, the TCPy levels observed in our Iranian sample were lower than the mean and 95th percentile (P95) levels set by HBM4EU-aligned studies [23]. Iran could benefit from a similar national database to better understand exposure across various populations and regions.

Several other studies assessed chlorpyrifos exposure by urinary levels in general populations. Berkowitz et al. and Meeker et al. evaluated chlorpyrifos levels in mothers and men, respectively [56,57], reporting higher concentrations than in our study, likely due to methodological and environmental differences. A notable study from Iran, in Rasht (part of the Mashhad supercluster), measured chlorpyrifos levels in maternal urine and breast milk, showing infant exposure risks [58]. Although minor, variations between this study and ours (1.3 µg/L in Rasht vs. 1.13 and 0.41 µg/L nationally and in Mashhad, respectively) may arise from differences in measurement techniques and populations.

**Table 8. Comparing the average concentration of Chlorpyrifos metabolite in the urine of Iranian adults with other similar studies.**

| Pesticide | Study | Country | Study Period | N | Population | Mean (RV95) | Adjusted Mean (Rv95) |
|---|---|---|---|---|---|---|---|
| Chlorpyrifos | Current study | Iran | 2019-2020 | 490 | General population | 0.75 (2.8) µg/L | 0.79 (2.9) µg/g |
| Chlorpyrifos | Dalsager et al. (2019) [47] | Denmark | 2010-2012 | 948 | Pregnant women | 2.08 (8.49) µg/L | – |
| Chlorpyrifos | Pirard et al. (2020) [48] | Belgium | 2016 | 229 | Children, 9–12 y | 4.92 (12.12) µg/L | – |
| Chlorpyrifos | Fernández et al. (2020) [49] | Spain (Valencia Region) | 2016 | 568 | Children, 5–12 y | 2.96 (11.08) µg/L | – |
| Chlorpyrifos | Namorado et al. (2020) [50] | Portugal | 2019-2020 | 296 | General population (Adults) | 7.35 µg/L | – |
| Chlorpyrifos | Probst-Hensch et al. (2020) [51] | Switzerland | 2020 | 299 | General population (Adults) | 3.64 µg/L | – |
| Chlorpyrifos | Weber et al. (2020) [52] | Germany | 2015-2020 | 180 | General population (Adults) | 2.87 µg/L | – |
| Chlorpyrifos | Eiríksdóttir et al. (2021) [53] | Iceland | 2019-2021 | 182 | General population (Adults) | 2.07 µg/L | |
| Chlorpyrifos | Govarts (2020) [54] | Belgium | 2019-2020 | 133 | General population (Children) | 3.24 µg/L | |
| Chlorpyrifos | Zock (2020) [55] | Netherlands | 2020 | 102 | General population (Children) | 3.49 µg/L | |
| Chlorpyrifos | Berkowitz et al. (2004) [56] | United States | 1998-2002 | 404 | General population (mothers and infants) | | 11.5 µg/g |
| Chlorpyrifos | Meeker et al. (2008) [57] | United States | 2000-2003 | N/A | General Population (men) | 2.59 µg/L | |
| Chlorpyrifos | Brahmand et al. (2019) [58] | Iran (Rasht) | 2017 | 61 | Mothers | 1.3 µg/L | |
| Chlorpyrifos | Galea et al. (2015) [59] | United Kingdom | 2011-2012 | 440 | Adults and children living near agricultural land | | 3.0 (9.6) µg/g |

The elevated chlorpyrifos levels in the Kermanshah supercluster have policy implications, emphasizing that targeted policies, rather than nationwide mandates, may be more effective. Regional dietary habits and agricultural practices likely influence these variations, as high dietary fat has been shown to increase chlorpyrifos bioavailability [63]. Chlorpyrifos has been banned in multiple countries, including parts of Europe and America [64], and Iran could consider similar restrictions. Additionally, the association between chlorpyrifos and visceral fat highlights the need for caution when assessing risk in populations potentially facing higher exposure. Nonetheless, further research is necessary to confirm these findings.

**Diazinon and malathion**

Diazinon and Malathion, two other pesticides assessed in this study, have been less researched compared to Chlorpyrifos. Diazinon is a moderately hazardous organophosphorus insecticide (class II, World Health Organization system) [65]. Limited studies on Diazinon exist; Garfitt et al., in a small study of five subjects, reported that urinary dialkyl phosphate levels peaked 2–12 hours after Diazinon exposure, with 60% of oral and 1% of dermal doses excreted as urinary DAP metabolites [66]. Malathion, another organophosphate, is used in mosquito-borne disease control and as a treatment for head lice [67,68]. The differences in urine levels of Diazinon and Malathion across the six Iranian superclusters indicate the need for region-specific policies on insecticide usage.

## Differences between men and women

This study examined the reported pesticides separately in men and women. For chlorpyrifos, men generally had higher corrected urinary levels; however, in Tehran, Zahedan, Mashhad, and Yazd, females exhibited higher levels. Malathion was slightly higher in men, while Diazinon was slightly lower. Among other demographic factors, significant associations were found between body muscle, body fat, and chlorpyrifos levels. A rat study linked chlorpyrifos exposure with weight gain [69], and the U.S. NHANES reported higher urinary concentrations of the pesticide 2,5-dichlorophenol in obese individuals [70].

## Strengths and limitations

This study provides valuable data on pesticide exposure in six representative regions in Iran, marking the first nationwide assessment of organophosphate pesticide exposure [25]. The study utilized the clustering model by Banfield et al. and Scrucca et al., producing a representative sample [29,30]. However, several limitations should be acknowledged. First, the urine levels reflect recent exposure, whereas long-term exposure may be more relevant for policy decisions. Second, comparing mean and RV95 urine pesticide levels across regions requires caution due to potential population differences. Third, the RV95 is a statistical model and not intended for health effect predictions, unlike human biomonitoring guidance values (HBM-GVs), which are essential for assessing population chemical exposure in health risk assessments [71]. This study did not establish HBM-GVs for the pesticides studied; future research should aim to determine these values. Additionally, lifestyle factors, diet, and other influential factors were not assessed, underscoring the need for further research. Despite clustering and modeling, the sample may not represent all populations across Iran's 31 provinces. The study's cross-sectional design, along with potential selection, information, and confounding biases, as well as self-reported data, may affect the generalizability of findings and prevent causal conclusions.

## Conclusion

In the current study, three main pesticides were measured in the urine of a nationwide sample, revealing several notable differences. Body visceral fat was found to influence urine pesticide metabolite levels. Additionally, non-zero levels were detected in 96.8%, 93%, and 65% of urine samples for Chlorpyrifos, Diazinon, and Malathion metabolites, respectively. Detectable amounts of these metabolites were present in the urine samples analyzed. The highest levels of Chlorpyrifos and Malathion were found in Kermanshah. Our findings offer a clear picture of pesticide exposure in Iran, based on a clustering method that represents the entire Iranian population. Given the extensive use of pesticides in Iran and the limited literature on the reported levels of these three main pesticides—Chlorpyrifos, Diazinon, and Malathion—our study provides regional reference levels and cutoffs that can help policymakers design programs to reduce pesticide exposure. Future research should focus on assessing additional pesticides and providing a more comprehensive evaluation of these organophosphates, with a larger sample size and the calculation of health-based guidance values [HBM-GVs].

## Supporting information

**S1 Table. Descriptive statistics on super-clusters (SCs), clusters and the assigned number of samples.**
(DOCX)

**S1 File. Urine pest.**
(RAR)

## Acknowledgments

We thank National Institute for Medical Research Development (NIMAD) Iran for financial support of project 964655.

## Author contributions

**Conceptualization:** Zahra Soleimani, Alireza Mesdaghinia, Farshad Farzadfar.

**Data curation:** Yousef Farzi.

**Formal analysis:** Yousef Farzi.

**Funding acquisition:** Alireza Mesdaghinia.

**Investigation:** Rosa Haghshenas, Zahra Soleimani, Amirmohammad Khalaji, Amir Hossein Behnoush, Ali Taherkhani, Ashkan Pourabhari Langroudi, Shahedeh Seyfi, Nasim Hadian, Mohammadhassan Momayyezi, Samaneh Mozafari, Samaneh Abolli.

**Methodology:** Rosa Haghshenas, Zahra Soleimani, Amirmohammad Khalaji, Amir Hossein Behnoush, Ali Taherkhani, Ashkan Pourabhari Langroudi, Shahedeh Seyfi, Nasim Hadian, Mohammadhassan Momayyezi, Samaneh Mozafari, Samaneh Abolli.

**Project administration:** Rosa Haghshenas, Zahra Soleimani.

**Resources:** Alireza Mesdaghinia.

**Supervision:** Kazem Naddafi, Masud Yunesian, Alireza Mesdaghinia, Farshad Farzadfar.

**Validation:** Ali Taherkhani, Nasim Hadian, Masud Yunesian, Farshad Farzadfar.

**Writing – original draft:** Rosa Haghshenas, Zahra Soleimani, Amirmohammad Khalaji, Masud Yunesian.

**Writing – review & editing:** Kazem Naddafi, Alireza Mesdaghinia, Farshad Farzadfar.

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
