## [Decision Letter · Decision Letter 0]

15 Aug 2024

Dear Dr. Mesdaghinia,

Thank you for submitting your manuscript to PLOS ONE. After careful consideration, we feel that it has merit but does not fully meet PLOS ONE’s publication criteria as it currently stands. Therefore, we invite you to submit a revised version of the manuscript that addresses the points raised during the review process.

We look forward to receiving your revised manuscript.

Kind regards,

Murtada D. Naser

Academic Editor

PLOS ONE

Journal Requirements:

5. We note that you have indicated that there are restrictions to data sharing for this study. For studies involving human research participant data or other sensitive data, we encourage authors to share de-identified or anonymized data. However, when data cannot be publicly shared for ethical reasons, we allow authors to make their data sets available upon request. For information on unacceptable data access restrictions, please see http://journals.plos.org/plosone/s/data-availability#loc-unacceptable-data-access-restrictions. 

Reviewers' comments:

Reviewer's Responses to Questions

**Comments to the Author**

1. Is the manuscript technically sound, and do the data support the conclusions?

Reviewer #1: Yes

Reviewer #2: Yes

Reviewer #3: Partly

Reviewer #4: Partly

Reviewer #5: Yes

Reviewer #6: Yes

2. Has the statistical analysis been performed appropriately and rigorously?

Reviewer #1: Yes

Reviewer #2: Yes

Reviewer #3: No

Reviewer #4: Yes

Reviewer #5: Yes

Reviewer #6: Yes

3. Have the authors made all data underlying the findings in their manuscript fully available?

Reviewer #1: Yes

Reviewer #2: Yes

Reviewer #3: No

Reviewer #4: No

Reviewer #5: Yes

Reviewer #6: Yes

4. Is the manuscript presented in an intelligible fashion and written in standard English?

Reviewer #1: No

Reviewer #2: Yes

Reviewer #3: No

Reviewer #4: No

Reviewer #5: Yes

Reviewer #6: Yes

Reviewer #1: The author may revise the manuscript with proper scientific evidence and supporting data to arrive at definite conclusions. The reasons are to be clearly defined and stated in the conclusions part.

Reviewer #2: Dear authors,

I read your manuscript"Reference Values of Urinary Metabolites of Organophosphate in Healthy Iranian Adults"

To improve the article's quality, I recommended the following comments:

1. In the materials and method section, the type of metabolites and their properties such as purity have not been written.

2. The name of the internal standard has not been mentioned. Please add the name of the internal standard along with a reference.

3. what do you mean by reference value? About reference value, I think that you should cite to new references. References of 17, 18, and 19 for this keyword are very old. What is the difference between reference dose and reference value?

4. Please cite new references for all materials and methods subtitles. Almost, more of them are without reference.

5. In the results section, numerical data lack unit. please check all of them.

6. All references are very old. I think it is better to discuss new references. The discussion section is written poorly.

7. Results are very poor. Except for Table 7( Visceral fat ), other results are without any significant relationship.

8. I think that the conclusion could be written better.

Best regards,

Reviewer #3: This manuscript presents a study on the urinary levels of organophosphate pesticide metabolites among a healthy Iranian adult population. The study provides valuable data on exposure levels across different regions and demographic groups within Iran. However, the manuscript requires significant revisions to meet the publication standards.

Title

Make the title more descriptive and specific about the study's focus, such as "Assessment of Organophosphorus Pesticide Exposure Among Iranian Adults Using Urinary Biomarkers."

Abstract

Abstract does not cover the main aim of the study which is derivation of reference values (RV95s). why this is not reported?

Ensure the abstract is concise and free from redundancy. Simplify complex sentences and use straightforward language to enhance readability.

Define any abbreviations or acronyms.

Clarify the results. For instance: Metabolite levels ranged from 0-24.9 µg/l for TCP (median: 2.8 µg/l, 2.9 µg/gcrt….

What 0 means in the range? Below detection limit? Please clarify.

Introduction

- The text adequately highlights the importance of the issue but could be more specific about the context of Iran (a more detailed explanation of the current state of pesticide use and exposure in Iran), and the gap in biomonitoring data specific to this region.

- Expand the literature review to include more recent studies and data on pesticide exposure and its health effects, particularly those relevant to the Iranian context.

- The introduction lacks citations in some areas, and some references are not formatted correctly (e.g., "(10())").

- Clearly state the study's objectives at the end of the introduction. This helps to provide a clear direction for the paper.

Please revise the line 115-119 as following:

“ In Iran, the use of pesticides has increased significantly over the last decade, raising serious concerns about their toxicological and detrimental effects. Despite this, there has been no national biomonitoring study or reported RV95 for pesticides in Iran. This study aims to fill this gap by calculating RV95s using data from an HBM study that measured pesticide metabolites in urine and examined effective demographic factors in Iran.”

The introduction lacks clear structure and flow, with some sentences being long and complex, making it difficult to follow. The transition between general information and the specific focus of the study is not smooth. Some examples:

Please revise the sentence line as: “The annual use of pesticides in agriculture continues to rise due to the expansion of agricultural activities”

And the following sentence as “ While pesticides are essential tools in modern agriculture for improving food supply, protecting crops, and controlling pests and disease vectors, their widespread use poses significant risks to human health and the environment”.

Material and method

Age range of the participants should be mentioned.

The methods section briefly mentions that "after data cleaning, 490 samples were used for statistical analysis." There are no details on the specific data cleaning procedures (checking for outliers, inconsistencies, duplicates, and invalid data entries) or criteria used to determine which samples were included or excluded (Samples with incomplete demographic data or urine samples with insufficient volume). How is the data cleaning process done? How did you handle the missing information? Describe the methods used for imputing missing data and justify the choice of imputation method.

Line 135-137: delete this sentence. It does not belong here.

Provide the durations at various temperatures to ensure sample integrity.

Information on validation procedures for analytical methods, including recovery rates, precision, accuracy, are missing.

I could not find use of quality control samples to monitor method performance. Please elaborate on descriptions of how calibration curves are validated and the specific acceptance criteria for QC sample

Line 165 remove “ 200”: The limit of detection 165 (LOD) and limit 200 of quantification (LOQ) for these compounds are indicated in table 2.

Line 182: what do you mean with “physical measurements” ? Anthropometry?

was any sensitivity analysis conducted to assess the impact of for example different cofounders or missing data on the study results?

Lots of grammatical error and long and complex sentences. Please revise the text.

Method for calculating of RV is missing.

While the importance of lifestyle factors, or food patterns or other factors in interpretation of the results is mentioned by the author in the introduction section but it seems no data on these aspects is collected in this study. This is one of the limitations of the study that should be mentioned in the discussion.

Results

The results are comprehensive but somewhat difficult to follow due to the dense text and lack of clear subsections. Clear headings and subheadings would improve readability.

The descriptive statistics for anthropometric measurements and body composition are well-presented. However, including more comparative statistics (e.g., t-tests or ANOVAs) could provide insight into whether these differences are statistically significant.

Line 200- revise as following:

“ The study population was predominantly female (58.09%). However, in the age group 55-65, the gender distribution was equal”

Revise line 204

“ Men were taller and weighed more than women, whereas women had higher BMI and body fat percentages. Detailed statistics are provided in Table 4.”

It is mentioned that the population in the study broken down into five quintiles, from the income/wealth ladder perspective. But the method used, and rationale is not mentioned in the method section. Question is why this approach is taken while relationship between pesticide concentrations in urine and this factor is not assesses.

Discussion:

The discussion disproportionately focuses on chlorpyrifos, with less detailed analysis of diazinon and malathion. A more balanced discussion would provide a comprehensive understanding of all three pesticides.

Discussion on the potential source of exposure to these pesticides is missing. Did author collect any data on lifestyle and dietary habits?

Occupational exposure to pesticide is excluded however information on the job of participants are missing.

The discussion of non-significant findings, especially regarding demographic factors, is lacking. Providing potential explanations or considerations for these results is recommended.

While comparisons with previous studies are made, more in-depth analysis and discussion of why differences might exist (e.g., differences in study populations, methodologies, and environmental factors) would be beneficial.

There is limited discussion on the potential biological mechanisms underlying the observed associations, particularly for the significant correlation between visceral fat and chlorpyrifos levels.

The discussion is somewhat disorganized, with some points repeated and others not clearly linked to the results. Clearer subheadings and a more structured approach would improve readability.

A more detailed evaluation of the study's limitations (including the cross-sectional design, potential biases in self-reported data, among others) and their potential impact on the results is required.

Discussion on RV95 is limited. More detailed information and comparison with other available RVs is required. are there any HBMGVs for these pesticides? please add more information on interpretation of the results.

Conclusion:

Expand the conclusion to discuss the broader public health implications of the findings and how they can inform future research and policy.

How this study filled the gap mentioned in the introduction?

Reviewer #4: The article reported metabolites of specific pesticides among 490 healthy Iranian adults to established reference values. Please, see my comments below.

1. The pesticides measured in this study include chlorpyrofos TCP, Diazinon, and Malathion. Are these four organophosphorus pesticides the most commonly used in Iran? In addition are there other pesticides and why are they not reported here. Furthermore, metabolites identified the only ones observed?

2. In the abstract the authors clearly state what the objective of the study is

3. #94: what is large quantities? Are you inferring that exposure to low levels do not safe?

4. There is the need to improve the language at several places in the text (e.g., #98-113)

5. #117: which definition?

6. #118-119 not clear

Methods

1. how can eligibility criteria help achieve representativeness. It's your sample strategy. Please, provide a brief description of your sample, sample size calculation, and data collection. I am concern about the eligibility criteria. I, For comparison among/between studies can you explain why adult was defined here as 25+ , (ii) the inclusion criteria not exposed to second hand smoke is a bit strange. How will you ensure this. Just inquiring about this with a questionnaire is not enough. Besides, it’s now established that combustion of biomass for cooking and heating produces similar chemicals as found in cigarette smoke. What about those exposed to household air pollution, (ii) what is your definition os healthy and how will you ensure this, (. iv) I find your eligibility criteria as too arbitrary. And also the one on occupational exposire

2. Provide detailed description on how the data was collected, by who and when?

3. The information provided on urine sampling is woefully inadequate

4. A bit of clarification is needed here about the exact instructions that were given to participants concerning the urine collection. The explanation should clarify for males and females.

5. Provide soe detail information about the questionniare and its administration in the text

6. provide a detail description of the analysis and the models that were used

Results

1. were females over-sampled? since the purpsoe of this study was to generate a reference for the iranian population, is the current population representative enough?

2. one of the serious flaws of this study was selection bias and not too sure if this study is suitable for the development of a reference

Discussions

1. discuss the reasons for the differences between communities. Else, it will appear as if the results are repeated

2. #291: what type of milk?

3. The authors should carefully discuss their results

4. The strength and limitations of thr study should be thoroughly discussed in the light of selection bias, information bias and confounding where possible

Reviewer #5: The study focuses on the metabolites of Chlorpyrifos (TCP), Diazinon (IMPY), and Malathion (Malathion dicarboxylic acid). Key parameters recorded include age, gender, wealth index, and body composition metrics such as body fat, muscle, visceral fat, and BMI. Urine samples were collected and analyzed using LC-MS/MS, and the data were processed through descriptive statistics and multiple linear regression.

However there are some limitations:

a.Sample Size and Representativeness: This study included 490 people from six provinces in Iran. Iran has 31 provinces with different geographical environments and agricultural distributions. Is the selected population representative?please explain.

b.Single-Time Urine Sample Collection: Urine samples were collected only once from each participant. Pesticide levels can fluctuate based on agricultural cycles, dietary changes, or seasonal variations.Urine samples should be collected three times on different days.

c.In disscusion,lack of recent references: please add the latest references and remodify discussion section.

d.The English in this article is not academic or standard enough. the language should be polished.

Reviewer #6: Abstract

authors provide urinary concentrations adjusted for creatinine. Authors should specify this so reader understands difference between concentrations in units of µg/L and µg/gcrt

Introduction

Line 94 – "when used in large quantities" – risk from pesticides is related to level of exposure and not quantities used.

Line 97 – different pesticides have different toxic properties. Authors should report toxic properties based on pesticide group (organophosphates, pyrethroids, etc) and not all pesticides together

General – introduction should include more information on status of registration of pesticides in Iran. Is chlorpyrifos approved for use on food crops? Do you have data on amount used? In EU for example this pesticide is no longer approved for plant protection.

Sampling

Was study population adults? Children? Although this appears in Table 1 authors should state in text that study population was adults

Were there questions in the questionnaire about diet? Distance of home from agricultural fields? If not this should be mentioned in limitations of study

Results

Lines 203-210 – findings of less interest to reader in context of this study

Line 244 – does this mean people with higher visceral fat had lower levels of TCPy?

Discussion

For readers to understand the findings, it is important to present study results in comparison to current studies in general populations from different countries worldwide. Table 8 includes mostly studies from the US, and many studies in occupational populations. I would encourage authors to look for more recent data emerging from HBM4EU study. Data is available on HBM4EU dashboard. Comparison of results for population in Iran should be compared to current studies (not from 1990s) and not from occupational studies.

Also, it is worth mentioning results in context of risk assessment. How do results compare to risk based HBM thresholds? See https://www.sciencedirect.com/science/article/pii/S143846392300010X#bib105

Line 295 – why do you think exposure levels were higher in Kermanshah region? Previous studies have shown that diet is major source of exposure to Chlorpyrifos. Is it possible that there are dietary differences between regions? Or do certain regions have more agriculture and possible drift exposure of the general population?

In policy discussion – it is worth mentioning that chlorpyrifos has been phased out in many countries, see recent decisions in EU and US.

**Do you want your identity to be public for this peer review?** For information about this choice, including consent withdrawal, please see our Privacy Policy

Reviewer #1: No

Reviewer #2: No

Reviewer #3: No

Reviewer #4: **Yes: ** Reginald Quansah

Reviewer #5: **Yes: ** Wan Long

Reviewer #6: No

---

## [Author Response · Author response to Decision Letter 1]

18 Dec 2024

Reviewer #1

The author may revise the manuscript with proper scientific evidence and supporting data to arrive at definite conclusions. The reasons are to be clearly defined and stated in the conclusions part.

Response: Thanks for this comment. As the reviewer asked correctly, we organized some sections of discussion and added subheadings for better understandability of the text in the discussion section.

Reviewer #2

Dear authors, I read your manuscript"Reference Values of Urinary Metabolites of Organophosphate in Healthy Iranian Adults"

To improve the article's quality, I recommended the following comments:

1. In the materials and method section, the type of metabolites and their properties such as purity have not been written.

Response: Thanks for this constructive comment by the reviewer. Corrected

2. The name of the internal standard has not been mentioned. Please add the name of the internal standard along with a reference.

Response: Thanks for your recommendation. We added it to the QA/QC section of the paper . Standard solutions were prepared using Pesticide Mix 1510 (Sigma Aldrich Company) at a concentration of 50 ppm, which included the pesticides Diazinon, Malathion, and Chlorpyrifos. From this mix, standards at concentrations of 20, 50, 100, and 200 ppb were prepared and analyzed in the LC-MS device in the same volume as the samples. Isotopically labeled Dialkyl Phosphates, including Diethyl Phosphate (DEP)-d10 and Dimethyl Phosphate (DMP)-d6, were used as internal standards.

3. What do you mean by reference value? About reference value, I think that you should cite to new references. References of 17, 18, and 19 for this keyword are very old. What is the difference between reference dose and reference value?

Response: Thank you for this comment. We have updated the references .The definition of the reference value of the German HBM Commission – RV95 – is the 95th percentile of the substance of interest at a specific time point rounded off within its 95% CI and agrees with the International Union of Pure and Applied Chemistry (IUPAC) guideline. The choice of the 95th percentile and 95% CI can be motivated by the convention in hypothesis testing where the 5% most extreme sample values indicate unusual values. But the A reference dose is “an estimate (with an uncertainty spanning perhaps an order of magnitude or greater) of a daily exposure level for the human population, including sensitive subpopulations, that is likely to be without an appreciable risk of deleterious effects” for chronic exposures.

4. Please cite new references for all materials and methods subtitles. Almost, more of them are without reference.

Response: Thanks for this constructive comment by the reviewer, we corrected this section according to your comments.

5. In the results section, numerical data lack unit. please check all of them.

Response: Thank you for this comment, Yes, we corrected that

6. All references are very old. I think it is better to discuss new references. The discussion section is written poorly.

Response: Thanks for this comment. As the reviewer asked correctly, we organized some sections of discussion and added subheadings for better understandability of the text in the discussion section.

7. Results are very poor. Except for Table 7( Visceral fat ), other results are without any significant relationship.

Response: Thank you for this comment. As this is a nationwide study, non-significant results are also important for policymakers and other researchers. Moreover, this study can pave the way for other researchers to know the effect of each variable on results. The main goal of this study was to evaluate reference values of urinary metabolites of organophosphate in healthy Iranian adults.

8. I think that the conclusion could be written better.

Response: We want to deeply thank the reviewer for this comment. We added better healthcare and policy implications of our findings in order to better elucidate the needs and future research.

Reviewer #3

This manuscript presents a study on the urinary levels of organophosphate pesticide metabolites among a healthy Iranian adult population. The study provides valuable data on exposure levels across different regions and demographic groups within Iran. However, the manuscript requires significant revisions to meet the publication standards.

Title

Make the title more descriptive and specific about the study's focus, such as "Assessment of Organophosphorus Pesticide Exposure Among Iranian Adults Using Urinary Biomarkers."

Abstract

Abstract does not cover the main aim of the study which is derivation of reference values (RV95s). why this is not reported? Ensure the abstract is concise and free from redundancy. Simplify complex sentences and use straightforward language to enhance readability.

Define any abbreviations or acronyms.

Clarify the results. For instance: Metabolite levels ranged from 0-24.9 µg/l for TCP (median: 2.8 µg/l, 2.9 µg/gcrt….

What 0 means in the range? Below detection limit? Please clarify.

Response: Thanks for this constructive comment by the reviewer. We corrected all points according to your comments.

Introduction

- The text adequately highlights the importance of the issue but could be more specific about the context of Iran (a more detailed explanation of the current state of pesticide use and exposure in Iran), and the gap in biomonitoring data specific to this region.

Response: Thanks for this constructive comment by the reviewer. We addressed this in the introduction.

- Expand the literature review to include more recent studies and data on pesticide exposure and its health effects, particularly those relevant to the Iranian context.

Response: Thanks for this constructive comment by the reviewer. We addressed this in the introduction.

The introduction lacks citations in some areas, and some references are not formatted correctly (e.g., "(10())").

Response: Thanks for this constructive comment by the reviewer. Yes, we corrected that.

- Clearly state the study's objectives at the end of the introduction. This helps to provide a clear direction for the paper.

Response:We want to deeply thank the reviewer for this comment. We added it to the last paragraph of the introduction

Please revise the line 115-119 as following:

“ In Iran, the use of pesticides has increased significantly over the last decade, raising serious concerns about their toxicological and detrimental effects. Despite this, there has been no national biomonitoring study or reported RV95 for pesticides in Iran. This study aims to fill this gap by calculating RV95s using data from an HBM study that measured pesticide metabolites in urine and examined effective demographic factors in Iran.”

Response: Thanks for this constructive comment by the reviewer .we changed the last paragraph to:

However, in Iran, despite a sharp increase in pesticide use over the last decade, there is a notable absence of national biomonitoring data and no reported RV95 values specific to the Iranian population. This gap in data raises significant concerns about the toxicological and health impacts of pesticide exposure in Iran. This study aims to address this critical gap by calculating RV95s for pesticide metabolites using data from an HBM study that analyzed urine samples and examined key demographic factors in Iran.

-The introduction lacks clear structure and flow, with some sentences being long and complex, making it difficult to follow. The transition between general information and the specific focus of the study is not smooth. Some examples:

Please revise the sentence line as: “The annual use of pesticides in agriculture continues to rise due to the expansion of agricultural activities”

And the following sentence as “ While pesticides are essential tools in modern agriculture for improving food supply, protecting crops, and controlling pests and disease vectors, their widespread use poses significant risks to human health and the environment”.

Response: Thanks for this constructive comment by the reviewer We revised the introduction section thoroughly and tried to address all the comments.

Material and method

Age range of the participants should be mentioned.

Response: Thank you for your comment. The age range of the participants in the study described in the article is 25 years or older. The age range is added to the Material and Method section.

The methods section briefly mentions that "after data cleaning, 490 samples were used for statistical analysis." There are no details on the specific data cleaning procedures (checking for outliers, inconsistencies, duplicates, and invalid data entries) or criteria used to determine which samples were included or excluded (Samples with incomplete demographic data or urine samples with insufficient volume). How is the data cleaning process done? How did you handle the missing information? Describe the methods used for imputing missing data and justify the choice of imputation method.

Response: The sampling design was developed based on a calculated sample size of 383, assuming a 50% baseline prevalence for urinary toxic metal(loid) levels, as no prior national surveys had assessed these parameters. Using Cochran's formula for large populations (Cochran 2007) and accounting for a 1.5 design effect (Salganik 2006) and an 85% response rate (Dey 1997), the sample size was adjusted to 660 across 132 clusters. Clusters were chosen via model-based clustering (Banfield and Raftery 1993; Scrucca et al., 2016), allowing us to organize Iranian provinces into six super-clusters (SCs) based on covariates such as particulate matter (PM10), pesticide use, and rates of cancer and chronic respiratory diseases. The capitals of these provinces—Ardabil, Kermanshah, Mashhad, Zahedan, Tehran, and Yazd—served as essential sampling units, with cluster coordinates within urban zones generated for systematic sampling (Bivand et al., 2019). Supplementary Table 1 provides details on each SC, cluster, and assigned sample size and inclusion and exclusion criteria.

Using the DIGIT data collection platform, ensured enough meta data to avoid duplication and data loss. This platform uses electronic questionnaires and multiple validation measures such as mandatory questions and limiting the answers to multiple choice to enhance data quality. An invalid data entry was impossible as the validations for each question and the answers were in place at the time of interviewing. However from 660 samples the ones without sufficient urine samples where excluded from data analysis (how many?).

The study incorporates non-response weights to adjust for participants who may not complete the full procedure (including questionnaires, anthropometry). There are nine adjustment weights to account for various types of non-responses, such as questionnaire-based non-response and laboratory-specific non-response for different sources (air, food, water, and urine). Any missing data in this was handled by using imputing average values for demographic data.

Line 135-137: delete this sentence. It does not belong here.

Provide the durations at various temperatures to ensure sample integrity.

Information on validation procedures for analytical methods, including recovery rates, precision, accuracy, are missing.

I could not find use of quality control samples to monitor method performance. Please elaborate on descriptions of how calibration curves are validated and the specific acceptance criteria for QC sample

Line 165 remove “ 200”: The limit of detection 165 (LOD) and limit 200 of quantification (LOQ) for these compounds are indicated in table 2.

Response: Thank you for your insightful question regarding the use of quality control (QC) samples in monitoring method performance. Calibration curves and quality control (QC) samples are essential components in validating analytical methods, particularly in LC-MS analysis of pesticides. A calibration curve establishes the relationship between known concentrations of an analyte and the instrument's response, allowing for the estimation of unknown concentrations in test samples. Calibration standards are prepared from a reliable stock solution, and the relationship is evaluated using linear regression analysis, with the correlation coefficient (r) ideally >0.995. According to regulatory guidelines, the accuracy of back-calculated concentrations from the calibration curve should be within ±15% of the nominal concentration for all levels except for the lower limit of quantification (LLOQ), which should be within ±20%, with at least 75% of the calibration standards meeting these criteria. QC samples are critical for monitoring the performance of the analytical method, with accuracy within ±15% of the nominal concentration for all levels except LLOQ, which should be within ±20%, and the coefficient of variation (CV) for QC samples at each concentration level ideally ≤15%, ensuring that an analytical run is only accepted if both calibrators and QCs meet their respective acceptance criteria.

Line 182: what do you mean with “physical measurements”? Anthropometry?

was any sensitivity analysis conducted to assess the impact of for example different cofounders or missing data on the study results?

Lots of grammatical error and long and complex sentences. Please revise the text.

Response: Thank you for your valuable comment. Physical measurements are height and weight plus assessing the visceral fat, muscle percentage, body fat. The physical measurements were assessed for having an impact on the results of the study. However, the findings were insignificant therefore not reported.

Method for calculating of RV is missing.

Response: Thank you for your comment. We explained about RV95 in the introduction and Material and Method section.

The entire manuscript was revised for grammatical errors and English structure.

Response: Thanks for this good comment by the reviewer. We revised the entire article in terms of writing

While the importance of lifestyle factors, or food patterns or other factors in interpretation of the results is mentioned by the author in the introduction section but it seems no data on these aspects is collected in this study. This is one of the limitations of the study that should be mentioned in the discussion.

Response: Thanks for this constructive comment by the reviewer. We added that to the limitation section, as the reviewer asked (highlighted): “Then, the effect of lifestyle factors, food patterns, and other influential factors were not assessed in the present investigation which highlights the need for further exploration of these associations.”

Results

The results are comprehensive but somewhat difficult to follow due to the dense text and lack of clear subsections. Clear headings and subheadings would improve readability.

Response: Thank you for this valuable comment which increased the readability of the results section. We added appropriate subheadings to the results section for better readability. The added subheadings are “Baseline characteristics,” “Anthropometric indices of men and women,” “Urine pesticide levels in the Iranian population,” and “The relationship between pesticide level and demographic features.”

The descriptive statistics for anthropometric measurements and body composition are well-presented. However, including more comparative statistics (e.g., t-tests or ANOVAs) could provide insight into whether these differences are statistically significant.

Response: I want to thank the respected reviewer for this great comment. The main goal of this study was to provide reference values for pesticide levels in the Iranian population. Thus, we focused our study on the association between anthropometric measurements and pesticide levels. To improve the readability and make the study more useful and concise, no statistical tests were conducted between different groups in anthropometric measurements.

Line 200- revise as

---

## [Decision Letter · Decision Letter 1]

4 Mar 2025

Dear Dr. Alireza Mesdaghinia,

We look forward to receiving your revised manuscript.

Kind regards,

Murtada D. Naser

Academic Editor

PLOS ONE

Journal Requirements:

Reviewers' comments:

Reviewer's Responses to Questions

**Comments to the Author**

Reviewer #2: All comments have been addressed

Reviewer #6: (No Response)

2. Is the manuscript technically sound, and do the data support the conclusions?

Reviewer #2: No

Reviewer #6: Yes

3. Has the statistical analysis been performed appropriately and rigorously?

Reviewer #2: No

Reviewer #6: Yes

4. Have the authors made all data underlying the findings in their manuscript fully available?

Reviewer #2: Yes

Reviewer #6: Yes

5. Is the manuscript presented in an intelligible fashion and written in standard English?

Reviewer #2: Yes

Reviewer #6: Yes

Reviewer #2: Dear authors,

please follow these comments:

1. In the abstract section, the following sentences should be placed after the aim of the article: "We measured metabolites for Chlorpyrifos (TCP: 2-isopropyl-4-methyl-6-hydroxypyrimidine), Diazinon (IMPY: 2-isopropyl-4-methyl-6-hydroxypyrimidine), and Malathion (Malathion dicarboxylic acid) in 490 healthy Iranian adults. Additionally, we recorded age, gender, wealth index, and body composition parameters including body fat, muscle mass, visceral fat, and BMI. Fasting urine sampling, along with body composition and demographic measurements, were conducted. Urine samples were subsequently analyzed."

2. What do you mean by "Our analysis included 300 ml of morning urine collected from participants on the fourth sampling day"?

3. Please correct Table 3, as the data presented is unclear. It is essential to apply appropriate statistical tests and analyses for comparing the data effectively.

4. In Tables 4, 5, and 6, please utilize a means comparison test to analyze the differences among the various variables, along with the corresponding P-values.

Reviewer #6: Authors should revise Table 8 to include more recent studies. Authors should consider removing occupational studies from this table. Authors mention HBM4EU study but should present data as well (perhaps in Table 8)

Line 325 - there have been studies showing toxicity of chlorpyrifos at levels lower than those that inhibit cholinesterase. Consider citing more recent studies here

Line 327 - "Studies in China and the U.S. reported higher pesticide poisoning rates in women than men" - this does not seem related to low level exposure (which is the topic of the current study) but perhaps suicide attempts? consider taking this statement out

**Do you want your identity to be public for this peer review?** For information about this choice, including consent withdrawal, please see our Privacy Policy

Reviewer #2: No

Reviewer #6: No

---

## [Author Response · Author response to Decision Letter 2]

30 Apr 2025

Manuscript number PONE-D-24-11112R2

Manuscript title: Reference Values of urinary metabolites of organophosphate in healthy Iranian adults

Dear Editor-in-chief,

Our team appreciates the editor and the reviewers for their observations and comments on the manuscript. We have fulfilled their comments and suggestions and wish to submit a revised version of the manuscript for further consideration in the journal. Changes in the revised version of the manuscript are highlighted in green color. Below, we also provide a point-by-point response explaining how we have addressed each of the reviewer and editors’ comments.

Yours sincerely,

Prof. Alireza Mesdaghinia

Review Comments to the Author

Please use the space provided to explain your answers to the questions above. You may also include additional comments for the author, including concerns about dual publication, research ethics, or publication ethics. (Please upload your review as an attachment if it exceeds 20,000 characters).

Reviewer #2:

1. In the abstract section, the following sentences should be placed after the aim of the article: "We measured metabolites for Chlorpyrifos (TCP: 2-isopropyl-4-methyl-6-hydroxypyrimidine), Diazinon (IMPY: 2-isopropyl-4-methyl-6-hydroxypyrimidine), and Malathion (Malathion dicarboxylic acid) in 490 healthy Iranian adults. Additionally, we recorded age, gender, wealth index, and body composition parameters including body fat, muscle mass, visceral fat, and BMI. Fasting urine sampling, along with body composition and demographic measurements, were conducted. Urine samples were subsequently analyzed."

Response: Thank you for your valid comment. This part is removed from the first paragraph and is added after the aim of the study.

2. What do you mean by "Our analysis included 300 ml of morning urine collected from participants on the fourth sampling day"?

Response: Thank you for pointing out the unclear sentence. Since this manuscript is part of a larger study on food, water, and air intake relation with pollutants in urine in the Iranian population, the sampling of intakes happened for three days and on the fourth day the urine sample was collected. This information is not relevant to this manuscript though. Therefore, the sentence is revised as “The urine sample of 300 ml fasting urine was collected from participants”

3. Please correct Table 3, as the data presented is unclear. It is essential to apply appropriate statistical tests and analyses for comparing the data effectively.

Response: Thank you for this comment, using the family assets questionnaires, a principal component analysis was implemented to create an index showing the wealth status of each family. The created index, then was categorized to 5 different levels from the poorest (Q1) to wealthiest (Q5). Table 3 shows the number of participants in each province and based on each of wealth level (Q1-Q5).

4. In Tables 4, 5, and 6, please utilize a means comparison test to analyze the differences among the various variables, along with the corresponding P-values.

Response: Thanks for this constructive comment by the reviewer. Corrected, the necessary explanations have been added to the text and tables.

Reviewer #6:

Authors should revise Table 8 to include studies that are more recent. Authors should consider removing occupational studies from this table. Authors mention HBM4EU study but should present data as well (perhaps in Table 8)

Response: We want to deeply thank the reviewer for this comment, as the referee rightly asked, we revised the table based on this comment and added the relevant studies to the table.

1. Dalsager L, Fage-Larsen B, Bilenberg N, Jensen TK, Nielsen F, Kyhl HB, Grandjean P, Andersen HR. Maternal urinary concentrations of pyrethroid and chlorpyrifos metabolites and attention deficit hyperactivity disorder (ADHD) symptoms in 2-4-year-old children from the Odense Child Cohort. Environ Res. 2019 Sep;176:108533. doi: 10.1016/j.envres.2019.108533. Epub 2019 Jun 11. PMID: 31229776.

2. Pirard C, Remy S, Giusti A, Champon L, Charlier C. Assessment of children's exposure to currently used pesticides in wallonia, Belgium. Toxicol Lett. 2020 Sep 1;329:1-11. doi: 10.1016/j.toxlet.2020.04.020. Epub 2020 May 1. PMID: 32371136.

3. Yusà V, F Fernández S, Dualde P, López A, Lacomba I, Coscollà C. Exposure to non-persistent pesticides in the Spanish population using biomonitoring: A review. Environ Res. 2022 Apr 1;205:112437. doi: 10.1016/j.envres.2021.112437. Epub 2021 Nov 26. PMID: 34838757.

4. Rodzaj W, Wileńska M, Klimowska A, Dziewirska E, Jurewicz J, Walczak-Jędrzejowska R, Słowikowska-Hilczer J, Hanke W, Wielgomas B. Concentrations of urinary biomarkers and predictors of exposure to pyrethroid insecticides in young, Polish, urban-dwelling men. Sci Total Environ. 2021 Jun 15;773:145666. doi: 10.1016/j.scitotenv.2021.145666. Epub 2021 Feb 6. PMID: 33596511.

5. Faniband MH, Norén E, Littorin M, Lindh CH. Human experimental exposure to glyphosate and biomonitoring of young Swedish adults. Int J Hyg Environ Health. 2021 Jan;231:113657. doi: 10.1016/j.ijheh.2020.113657. Epub 2020 Oct 30. PMID: 33130428.

Line 325 - there have been studies showing toxicity of chlorpyrifos at levels lower than those do that inhibit cholinesterase. Consider citing more recent studies here

Response: Thank you for your comment. Line 324, updated references are added to the text.

Burke RD, Todd SW, Lumsden E, Mullins RJ, Mamczarz J, Fawcett WP, Gullapalli RP, Randall WR, Pereira EFR, Albuquerque EX. Developmental neurotoxicity of the organophosphorus insecticide chlorpyrifos: from clinical findings to preclinical models and potential mechanisms. J Neurochem. 2017 Aug;142 Suppl 2(Suppl 2):162-177. doi: 10.1111/jnc.14077. PMID: 28791702; PMCID: PMC5673499.

Wołejko E, Łozowicka B, Jabłońska-Trypuć A, Pietruszyńska M, Wydro U. Chlorpyrifos Occurrence and Toxicological Risk Assessment: A Review. Int J Environ Res Public Health. 2022 Sep 26;19(19):12209. doi: 10.3390/ijerph191912209. PMID: 36231509; PMCID: PMC9566616.

Line 327 - "Studies in China and the U.S. reported higher pesticide poisoning rates in women than men" - this does not seem related to low level exposure (which is the topic of the current study) but perhaps suicide attempts? consider taking this statement out

Response: Thank you for raising this valid comment in the line 372. The statement about higher pesticide poisoning rates in women from studies in China and the U.S. does appear to be about acute pesticide poisoning, is a different issue that often involves intentional exposure or occupational accidents. We omitted the sentence from our text.

---

## [Editor Report · Decision Letter 2]

7 May 2025

Reference Values of urinary metabolites of organophosphate in healthy Iranian adults

PONE-D-24-11112R2

Dear Dr. Alireza Mesdaghinia,

We’re pleased to inform you that your manuscript has been judged scientifically suitable for publication and will be formally accepted for publication once it meets all outstanding technical requirements.

Kind regards,

Murtada D. Naser

Academic Editor

PLOS ONE
---

## [Editor Report · Acceptance letter]

PONE-D-24-11112R2

PLOS ONE

Dear Dr. Mesdaghinia,

I'm pleased to inform you that your manuscript has been deemed suitable for publication in PLOS ONE. Congratulations! Your manuscript is now being handed over to our production team.

Kind regards,

on behalf of

Dr. Murtada D. Naser

Academic Editor

PLOS ONE